# Biomimetic Sensors to Detect Bioanalytes in Real-Life Samples Using Molecularly Imprinted Polymers: A Review

**DOI:** 10.3390/s21165550

**Published:** 2021-08-18

**Authors:** Birgit Bräuer, Christine Unger, Martin Werner, Peter A. Lieberzeit

**Affiliations:** Department of Physical Chemistry, Faculty for Chemistry, University of Vienna, Waehringer Strasse 42, 1090 Vienna, Austria; birgit.braeuer@univie.ac.at (B.B.); christine_unger@univie.ac.at (C.U.); martin.werner@univie.ac.at (M.W.)

**Keywords:** molecularly imprinted polymers, biomimetic sensing, real-life matrices, diagnostic applications, environmental sensing, food safety, illicit substances

## Abstract

Molecularly imprinted polymers (MIPs) come with the promise to be highly versatile, useful artificial receptors for sensing a wide variety of analytes. Despite a very large body of literature on imprinting, the number of papers addressing real-life biological samples and analytes is somewhat limited. Furthermore, the topic of MIP-based sensor design is still, rather, in the research stage and lacks wide-spread commercialization. This review summarizes recent advances of MIP-based sensors targeting biological species. It covers systems that are potentially interesting in medical applications/diagnostics, in detecting illicit substances, environmental analysis, and in the quality control of food. The main emphasis is placed on work that demonstrates application in real-life matrices, including those that are diluted in a reasonable manner. Hence, it does not restrict itself to the transducer type, but focusses on both materials and analytical tasks.

## 1. Introduction

The first report on molecular imprinting appeared in 1931, when M. V. Polyakov found the unusual adsorption behavior of silica particles that were prepared in the presence of soluble additives [1]. Molecular imprinting describes a process to synthesize polymers as artificial receptors for a desired target analyte. The target species acts as a template, which initially forms a complex with functional monomers through covalent or non-covalent interactions. Polymerization in the presence of a crosslinker and subsequent removal of the target molecule lead to template-specific cavities regarding shape and functionality that are capable of rebinding the target analyte selectively (Figure 1) [2].

Dependent on the application, MIPs come in different shapes, such as monoliths, thin films, or nanoparticles [3]. Synthesizing MIPs is possible in various ways, including free radical polymerization [4], electropolymerization [5,6], controlled/living polymerization [7], or sol gel synthesis [8] for a wide range of analytes, starting from ions [9] up to whole cells [10]. There are reports of numerous applications of MIPs, e.g., solid-phase extraction [11,12], catalysis [13,14,15], drug delivery [16,17], or sensing [18]. When detecting a specific analyte, the respective selective receptor is the core piece of the sensing set-up. MIPs as receptor elements come with the advantage of high stability towards environmental conditions, and a low cost compared to biological recognition elements. Furthermore, the production is versatile and can be adjusted easily towards the respective target analyte. There are many examples for MIPs used for sensing [19,20,21]. For actual commercialization, it is important to access the ability of the MIPs to perform in real samples. The basic concept of MIPs in sensing set-ups needs to prove its ability to work in complex matrices and not just in controlled environments. This review gives an overview over MIPs used in the detection of analytes in real samples for medical applications, food safety, environmental monitoring, and detecting illegal substances. We discuss sensor performance with a focus on application in complex samples, especially concerning sample pre-treatment.

## 2. Medical Applications

Biosensing platforms developed for application in medicine aim at detecting very diverse analytes ranging from small molecules (e.g., in therapeutic drug monitoring [22]), or proteins and other biomolecules (e.g., as biomarkers for early detection of diseases [23,24]), to whole cells for pathogen identification and diagnosis [25,26]. The applicability of the sensors for measurements in complex biological matrices, such as blood serum or whole blood, urine, saliva, and other biological fluids, is crucial for their real-life use in clinical environments. Hence, such sensors are required to offer sufficient selectivity and sensitivity towards the target analyte with minimum cross-selectivity towards other constituents of biological matrices. A vast number of such sensing systems have been developed based on molecularly imprinted polymers as biomimetic entities for target recognition. The following sections discuss MIP-based sensors designed for clinical purposes with regard to their imprinting strategies and potential for application in real-life clinical matrices.

### 2.1. Prevention and (Early) Diagnosis of Diseases

The term “biomarker” refers to measurable species that are involved in physiological or biological processes and can be associated with a certain diagnosis depending on their presence, concentration, structure, function, or fluctuation. Tracking such biomarkers in complex biological matrices (e.g., blood, urine or saliva) is important for early diagnosis, for instance of cancer and cardiovascular diseases, as well as disease monitoring following treatment. Biomarkers are often present in trace amounts in complex biological matrices, which poses substantial challenges for their sensing in terms of sufficient sensitivity and selectivity [27].

Cancer is among the leading causes of death. Hence, detecting cancer biomarkers at an early stage is crucial for successful treatment and improved prognosis for patients [27]. Pirzada et al. [28] recently reported a MIP-based sensing system for cancer biomarker detection. Their concept is based on hybrid epitope imprinting combined with gold nanoparticles for signal amplification to detect neuron specific enolase (NSE), a small-cell lung cancer biomarker. They used square wave voltammetry to detect their target. Epitope imprinting is a favorable approach for protein imprinting, which uses only a selected part of the protein surface—the epitope—as the template, which makes production a lot cheaper. The authors chose two cysteine-modified epitopes of NSE, adsorbed them to a gold electrode and fabricated the corresponding MIP by electropolymerization of scopoletin as the functional monomer in the presence of gold nanoparticles (Figure 2). They observed linear detection for NSE between 50 and 500 pg/mL in PBS buffer, and between 25 and 500 pg/mL in human serum diluted 1:1 with PBS buffer. Compared with MIPs fabricated in the same manner, but without gold nanoparticles, revealed that the latter increased sensitivity by a factor of 8 and binding affinity by a factor of more than 3.5. The study demonstrates that hybrid epitope imprinting, in combination with nanomaterial amplification, is feasible for sensitive protein biomarker detection, even in complex matrices such as human serum.

Horikawa et al. [29] developed another approach to fabricate highly sensitive and selective MIPs for protein detection using α-fetoprotein (AFP), a biomarker for liver cancers. Their strategy involves post-imprinting modification (PIM) following covalent protein imprinting: the authors covalently attached two cleavable functional monomers to AFP and, following polymerization, introduced two PIMs specifically into the AFP-imprints in a multi-step process. While the first PIM was a multifunctional entity specific for AFP binding, the second served the purpose of transducing AFP binding into a change in fluorescence. AFP binding evaluated by fluorescence microscopy yielded an affinity constant (Ka) of 1.5 × 10^10^ M^−1^ in 10 mM phosphate buffer, which was not significantly changed for measurements in 1% (Ka 1.5 × 10^10^ M^−1^) and 10% (Ka 1.4 × 10^10^ M^−1^) human serum. Moreover, selectivity of the MIP towards AFP was investigated by evaluating their responses to human serum albumin and prostate-specific antigen, which turned out negligible.

Mori et al. [30] proposed a similar approach to MIP-based sensing in complex matrices involving PIMs for the fluorescence-mediated detection of exosomes relevant to prostate cancer in teardrops. Imprinting of exosomes originating from the human prostate-cancer cell line PC3 took place via surface-initiated atom-transfer radical polymerization (ATRP) using 2-methacryloyloxyethyl phosphorylcholine as a monomer. Following removal of the template, authors specifically introduced both an antibody targeting CD9 (a protein expressed on most exosomes) for exosome binding and fluorescence reporter molecules to detect target binding into the imprints. The system turned out highly sensitive to exosomes (LOD 6 pg/mL). In order to differentiate cancer-related exosomes from normal exosomes, MIPs were prepared with antibodies for CD9 or GGT1 (an exosomal protein overexpressed on prostate-cancer cell exosomes). The authors demonstrated that normal exosomes showed different responses to the MIPs compared to those secreted by the PC3 cell line, thus allowing for differentiating them. Moreover, they demonstrated successful sensing of exosomes from teardrops without sample pre-treatment. In summary, both publications indicate that PIMs are a powerful tool to increase sensitivity and selectivity of MIPs and render them suitable for application in complex clinical matrices.

Cardiovascular diseases are the leading cause of death worldwide, accounting for about 31% of global deaths with most fatalities associated with coronary heart disease and stroke. Thus, early detection of cardiovascular diseases has become increasingly important, often through sensing of corresponding biomarkers. For that purpose, biosensors offer a rapid, accurate, and specific detection platform as an alternative to conventional time-consuming methods [27]. Various biomarkers have been identified for this purpose, such as very-low-density lipoprotein [31], low-density lipoprotein [32], and high-density lipoprotein [33].

Chunta et al. [34], for instance, published a concept for MIP-based detection of oxidized-low-density lipoprotein (oxLDL). This compound is patho-atherogenic; elevated serum levels indicate increased risk for atherosclerosis, resulting in coronary artery disease. The approach utilizes acrylate-based MIP thin films on quartz crystal microbalance (QCM). The authors demonstrated that the sensing system could be used to detect ox(LDL) in a concentration range between 86 and 5600 µg/dL with an LOD of 86 µg/dL and an LOQ of 287 µg/dL in 10 mM PBS (pH 7.4). This makes the sensor suitable for measuring ox(LDL) concentrations corresponding to both clinically normal and pathological values. Selectivity investigations of the sensor with low-density lipoprotein (LDL), high-density lipoprotein (HDL), very-low-density-lipoprotein (VLDL), and human serum albumin (HSA) revealed that those competitors led to signals of about 10% or lower compared to the signal for oxLDL, indicating sufficient selectivity (Figure 3). Moreover, binding oxLDL in artificial serum (diluted 1:2 with 10 mM PBS (pH 7.4)) turned out successful. The sensor responses were in good agreement with the results obtained from a conventional ELISA assay. Similar MIP-based approaches were published by the same group for the sensing of HDL [33], VLDL [31], and LDL [32] in human blood serum.

Among biomarkers for early detection of diseases, HSA is one of the most popular targets for sensing: elevated levels of this protein in urine (termed microalbuminuria) are an early sign of kidney malfunction. Moreover, microalbuminuria is also correlated with heightened cardiovascular morbidity, which emphasizes its importance in detecting renal and cardiovascular disorders [35]. Stojanovic et al. [35] published an approach to HSA detection involving HSA-MIP fabrication via electropolymerization of scopoletin on gold disk electrodes in the presence of HSA in combination with cyclic voltammetry to investigate target rebinding. The sensing system showed a linear response behavior for concentrations between 20 and 100 mg/L with an LOD of 3.7 mg/L. Expected concentrations for HSA in blood are between 35 and 50 g/L, and urine levels for healthy individuals are up to 25 mg/L, compared to up to 200 mg/L for microalbuminuria. As the sensor exhibited an operating range between 20 and 300 mg/L, the authors expect it to be applicable to real-life urine samples. In order to test this hypothesis, urine samples collected from healthy and diabetic volunteers were analyzed. HSA concentrations were compared to a standard immunoturbidimetric technique. The deviation between the standard method and the MIP-based sensor for the samples from diabetic patients was found to be less than 10%, which the authors found acceptable considering general uncertainties associated with HSA determination. However, in the sample donated by the healthy individual, HSA concentration was found to be below the limit of detection. Therefore, sensitivity needs further improvement in order to ensure that the method is fit to analyze real-life clinical samples.

Zhang et al. [36] published another approach to HSA sensing based on electropolymerization using o-phenylenediamine and hydroquinone as functional monomers. They provided a method for enhanced sensor sensitivity by functionalizing a gold electrode with gold nanoparticles, and polythionine-methylene blue as electrocatalytic material. HSA binding changed signals in a differential pulse voltammetry using a dual-signal method, with polythionine-methylene blue as a substrate redox probe and [Fe(CN)_6_]^3−/4−^ as a solution redox probe. The response behavior of the sensing platform was linear at concentrations between 1.0 × 10^−10^ and 1.0 × 10^−4^ g/L in 0.1 mol/L PBS (pH 6.0) with an LOD of 3.0 × 10^−11^ g/L. Additionally, urine samples of one healthy and one ill volunteer revealed that HSA levels were consistent with the results obtained from the immunoturbidimetric standard method.

The past few years have seen a wide range of methods aimed at sensing diagnostic and prognostic biomarkers in real-life clinical matrices such as blood and urine. Table 1 covers a selection of those, including analyte and characteristic parameters.

### 2.2. Pathogen Detection

Rapid detection and identification of pathogens is essential to control and prevent the spread of infectious diseases, for accurate diagnosis, and to optimize treatment. Standard methods of pathogen detection comprise microscopy, immunoassays, culturing of pathogens, and nucleic acid amplification. All of these techniques require specifically trained staff and are rather time-consuming. Again, sensors based on molecularly imprinted polymers provide a fast and straightforward alternative to detect such pathogens, e.g., bacteria, viruses, fungi, and parasites [27].

The Hepatitis A virus (HAV) is associated with infectious liver disease, infecting one to two million persons globally every year. The fatality rate of Hepatitis A strongly depends on age and ranges from 0.1–0.3% in children to 1.8–5.4% in the age group 50+ [47]. Luo et al. [48] published a sensing platform for HAV based on MIP nanoprobes fabricated from a pH-responsive polymer and MIL-101 as a substrate material using Resonance Light Scattering to detect template rebinding. The metal-organic framework substrate MIL 101 exhibits a large surface area, which improves the sensitivity and linear range of the nanoprobes. This results in a linear response behavior for HAV concentrations between 0.02 and 2 nmol/L in phosphate buffer, a detection limit of 0.1 pmol/L, and a short response time of 20 min. Measurements in human serum diluted 200-fold with phosphate buffer and spiked with HAV concentrations ranging from 0.1 to 1.5 nmol/L revealed recovery rates between 88% and 108%.

Human immune deficiency virus (HIV) is the pathogen associated with acquired immune deficiency syndrome (AIDS), which is still incurable. HIV-1 is the most widely spread type of HIV. Hence, it is particularly interesting for the diagnosis and monitoring of the HIV infection, as well as for controlling the spread of the virus. Since HIV-p24, the capsid protein of HIV-1, is present in infected individuals prior to antibody development, it is more useful for early detection of HIV infection [49] than the latter. Ma et al. [49] constructed MIPs for the detection of HIV-p24 based on a glassy carbon electrode modified with multi-walled carbon nanotubes, which were equipped with a HIV-p24 MIP by surface polymerization utilizing acrylamide as a functional monomer and *N*,*N*′-methylene bis-acrylamide as a crosslinker. The multi-walled carbon nanotubes served the purpose of increasing the specific surface area, electrocatalytic performance, and electrical conductivity, which enhances sensitivity. Differential pulse voltammetry in PBS confirmed a linearly increasing response for HIV-p24 concentrations between 1.0 × 10^−4^–0.01 ng/mL and 0.01–2 ng/mL for high and low affinity binding sites, respectively, with an LOD of 0.083 pg/mL. Tests in spiked human serum samples led to recovery rates between 98.2% and 100.3%, in good agreement with the standard ELISA method.

Japanese Encephalitis Virus (JEV) is a mosquito-borne virus that is responsible for Japanese Encephalitis, a neurotrophic disease with a case fatality rate between 5% and 30% [50]. Liang et al. [51] published a technique for JEV detection using fluorescent-modified silica microspheres as a support for a JEV-imprinted polymeric layer fabricated with APTES as a functional monomer and TEOS as a crosslinker. Binding of the virus to the MIP particles was monitored utilizing fluorescence resonance energy transfer (FRET), where the virus served as an energy donor and the fluorescent dye pyrene-1-carboxaldehyde (PC) as an energy acceptor. In ultrapure water, the sensor exhibited a linear response behavior between 24 and 960 pM, with an LOD of 9.6 pM. Applicability of the sensing strategy to real-life samples was assessed by detection of JEV in human serum samples diluted 2000-fold with deionized water and the spiking of JEV concentrations between 50 and 500 pM. Recovery rates were between 97.7% and 100.5%, indicating the successful sensing of JEV. However, it has to be emphasized that the human serum was diluted 2000-fold prior to analysis, and the results do not apply to undiluted human serum.

Besides viruses, bacteria are also of fundamental interest in sensing: bacteria-borne infections range from common mild illnesses to severe and fatal diseases [52]. Steen Redeker et al. [52] introduced a method to identify nine different bacteria species utilizing surface imprinted polyurethane in combination with thermal heat transfer resistance measurements. The corresponding sensor responses were dependent on bacteria concentration. In PBS, the sensor setup achieved an LOD of 1.01 × 10^4^ CFU/mL for *E. coli*, and between 1 × 10^5^ and 2 × 10^5^ CFU/mL for the other bacteria species. Selectivity experiments revealed appreciable selectivity, except for two *E. coli* strains due to their similarity (Figure 4). To ensure real-life detection, the authors measured a clear urine sample of a healthy person spiked with different *E. coli* concentrations. It turned out possible to quantify *E. coli*, but the LOD in urine increased to 3 × 10^4^ CFU/mL, which the authors attribute to the vast amount of proteins and other interferents present in urine.

Detecting bacteria on contaminated surfaces can help in preventing infections in clinical environments. Van Grinsven et al. [53] reported a similar method for detection of *E. coli* originating from contaminated surfaces based on *E. coli*-imprinted polyurethane layers on aluminum chips in combination with a thermal readout technique. In order to assess the applicability of the method to real-life detection of bacterial surface contamination, the authors contaminated lab benches in a controlled manner, ensuring that only the target bacteria were present. The detection limit in the swab rinse kit (SRK) buffer was found to be in the low 10^4^ CFU/mL range; the dynamic range was about one order of magnitude. *E. coli* concentrations obtained from the swabs were determined using the MIP/heat-transfer method and compared to results obtained from conventional bacteria culturing and colony-counting. The resulting recovery rates were between 16% and 65%. Overall, the results indicate that the method is suitable for the detection of *E. coli* from surfaces, but the dynamic range is rather small. Thus, it is necessary to enhance the binding capacity of the recognition layer to render the device applicable to real-life sensing.

### 2.3. Therapeutic Drug Monitoring

Safe and successful disease treatment and drug efficacy optimization require monitoring of actual concentrations of prescription drugs in biological matrices, mostly in blood serum or plasma [54]. Conventional methods for therapeutic drug monitoring include immunoassays, approaches based on separation techniques, such as high-performance liquid chromatography (HPLC), mass spectrometry, electrochemistry, and chemiluminescence [22,55]. While they yield sufficiently accurate concentration results, they are time-consuming and rely on expensive instrumentation, elaborate sample preparation, and qualified staff [55]. In contrast, biosensors based on molecularly imprinted polymers are thought to provide a way for a low-cost, rapid, and accurate analysis with minimal sample preparation [22].

In the context of MIP-based sensors in therapeutic drug monitoring, paracetamol is a particularly popular target as it is a commonly used non-prescription drug. On the other hand, overdosing it can result in severe side effects [55]. Alanazi et al. [55] recently established rapid and sensitive paracetamol sensing based on electroactive MIP nanoparticles. The nanoMIPs were fabricated by solid phase synthesis using paracetamol-functionalized glass beads in the presence of itaconic acid as a functional monomer; for sensing, immobilized MIP nanoparticles on carbon electrodes were used for differential pulse voltammetry. To adapt the sensing system to clinically relevant paracetamol concentrations for the prevention of kidney and liver failure (0.1–1 mM), the authors adjusted the concentration of the immobilized nanoMIPs on the electrode, accordingly, in a successful manner. Optimization led to a sensor with acceptable linearity in the relevant concentration range (0.1–1 mM), LOD and LOQ of 50 and 167 µM, respectively, and a sensitivity of 6.18 ± 0.22 μA/mM in spiked human plasma. Furthermore, selectivity assessment of the sensing system in PBS buffer showed that other drugs including caffeine, procainamide, and ethyl 4-aminobenzoate showed no interaction.

Another example for an MIP-based sensing system targeting paracetamol in human serum samples was published by Yarman et al. [56], who electropolymerized a mixture of o-phenylenediamine and resorcinol as functional monomers in the presence of paracetamol to obtain MIP-covered glassy carbon electrodes for amperometric measurements at a low electrode potential (−0.1 V). This was facilitated by the treatment of paracetamol with tyrosinase, yielding N-acetyl-p-benzoquinone, which gave a cathodic current signal at −0.1 V. In 50 mM phosphate buffer (pH 7), no interfering signals were observed for ascorbic and uric acid, and only 20 percent of the paracetamol signal for levodopa and catechol. For paracetamol concentrations between 2.5 µM and 335 µM, the sensor response showed linearity. However, as commonly used paracetamol dosages result in blood concentrations between 0.2 and 2 mM, the authors had to dilute the serum by a factor of 1:10 to 1:100 in order to measure in the linear range of the sensor. In addition, they observed a decrease in current signal of 30 percent in paracetamol spiked diluted human serum compared to measurements in the buffer, which they attributed to serum protein adsorption on the electrode surface, reducing accessibility for the analyte.

Serum proteins are found in human blood serum in the millimolar concentration range and, therefore, disturb sensing of protein biomarkers or drugs, which are typically present at nanomolar concentrations. Selectivity of MIPs is often not sufficient to outweigh this excess of serum proteins [57]. Thus, MIPs aimed at the sensing of proteins in human serum are usually evaluated in deproteinated or semi-synthetic serum. For instance, Ozcelikay et al. [57] developed a MIP-based sensor for monitoring the antibiotic lipopeptide daptomycin in deproteinated human serum. They generated the MIP by electropolymerization of o-phenylenediamine in the presence of daptomycin on a glassy carbon electrode modified with Au-Pt nanoparticles, and monitored template removal and rebinding by cyclic voltammetry (Figure 5) and differential pulse voltammetry. In deproteinated human serum samples, the sensor showed a linearly increasing response in a concentration range between 1 and 50 pM. Moreover, they assessed the selectivity of the sensor against Vancomycin and Erythromycin (antibiotics very similar to DAP in structure) as well as glycin and tryptophan, amino acids which are part of the DAP primary structure. The sensitivity of the sensing system was below seven percent for all of those species. Although the sensor showed high selectivity and sensitivity towards daptomycin, sample pre-treatment, namely serum deproteination and dilution, is a drawback associated with this sensor, and many MIP-based approaches to sensing of not only proteins but also small molecules [58,59].

Theophylline is an inexpensive drug, which is commonly prescribed under several brand names to treat diseases of the respiratory system, such as asthma or chronic obstructive pulmonary disease. Due to the observation that adverse reactions are associated with higher plasma concentrations (above 110 µM), monitoring of the drug in plasma is crucial for treatment safety. Gan et al. [60] developed an electrochemical MIP sensor for theophylline detection based on a core-shell structured SiO_2_@TiO_2_-embedded theophylline-imprinted polymer with methacrylic acid as a functional monomer. The authors reported no interference with theophylline signal for ions present in real samples (1000-fold concentration of Mg^2+^, Ca^2+^, Na^+^, K^+^, NH_4_^+^, Cl^−^, F^−^, CO_3_^2−^, SO_4_^2−^, 300-fold concentration of glucose, urea and oxalate, 10-fold concentration of uric and ascorbic acid as well as dopamine, measurements in 0.1 M PBS) or for structurally similar molecules such as caffeine, theobromine, xanthine, guanine, adenine, and 1,7-dimethyl xanthine (Figure 6). Moreover, the sensor was applied to human blood serum and urine without sample pre-treatment, and showed acceptable recoveries of 94.5% and 101%, respectively. The authors attribute the sensor’s high sensitivity and selectivity to the SiO_2_@TiO_2_ core structure, which leads to an increased active surface area.

MIP-based sensing platforms for therapeutic drug monitoring in complex clinical matrices have been published for many target analytes within the past few years. Table 2 presents a selection indicating respective linear concentration range, LOD, and type of matrix.

## 3. Food Safety

It is a worldwide challenge to provide the population with food that can be safely consumed. Typical hazards comprise both biological contaminations or chemical ones. Both require regulation and monitoring to prevent spreading of foodborne diseases. The use of chemical sensors for such applications is of interest, as they provide a cheap and easy-to-use alternative to the established, often time consuming and expensive analysis methods. Molecular imprinting is a powerful tool to help detect a large variety of analytes, including those relevant to food safety monitoring [66]. While many papers focus on developing such biomimetic sensors for various templates on different transducer types [66], the most important step in commercializing them is to prove their applicability for sensing in real food sample matrices. The following sections collect some examples that have demonstrated this for different analyte types.

### 3.1. Mycotoxins

Monitoring food contamination by mycotoxin-producing fungi is an important aspect of food safety, because the released toxic compounds are severely dangerous for humans. One possibility to detect such food contamination is to develop devices that can sense the mycotoxins themselves [67]. In the work of Sergeyeva et al. [68] a fluorescence sensor based on an in-silico-optimized MIP membrane was fabricated to detect aflatoxin B1 (AFB1), which is one of the most toxic mycotoxins produced by the fungal species *Aspergillus parasiticus* and *Aspergillus flavus*. It typically contaminates maize and wheat. Hence, the authors in one part of their work demonstrated the detection of AFB1 in maize extracts. Upon binding the analyte, the MIP membranes developed feature fluorescence emission after irradiation with UV light. This effect is accessible both via spectro-fluorimetry, and with the camera of a smartphone and a commercially available software application. Spectro-fluorimetry demonstrated a linear range of 15–500 ng/mL and an LOD of 15 ng/mL, whereas the performance slightly decreased when using smartphone detection, as only a linear range between 20 and 100 ng/mL, and an LOD of 20 ng/mL, could be obtained this way. Nevertheless, the use of a smartphone as a signal transducer and readout system has many advantages: among others, it is cheap and easy to use. In real-life matrices—i.e., extracts from maize samples—the sensor is able to detect aflatoxin B1 with a recovery rate 87% ± 7% and a recovery rate of at least 96% for the spiked samples (Table 3).

In a following study of Sergeyeva et al. [69], they reported on a fluorescence sensor based on an MIP membrane for the detection of zearalenone, another dangerous mycotoxin that is a potential contaminant of cereals. The MIP membrane-sensing layer was synthesized in a so-called “dummy-template”-based approach, which uses a structurally closely related template (i.e., cyclododecyl-2,4-dihydroxybenzoate) instead of the analyte for imprinting. As the dummy template is non-fluorescent, the incompletely removed template does not interfere with the measurements. Fluorimetric detection revealed a linear range of 1–25 µg/mL and an LOD of 1 µg/mL for zearalenone measurements. Six other mycotoxins led to only low cross-selectivity for this MIP. Only α-zearalenol—a metabolite of zearalenone—showed high cross-selectivity for this sensor. Again, the authors also tested the performance of the fluorescence sensor for their smartphone readout system, revealing a linear range of 1–10 µg/mL and an LOD of 1 µg/mL (Figure 7). Real-life samples including maize, wheat, and rye flour extracts (extracted in acetonitrile water) that were either spiked with 1–5 µg/mL zearalenone, or contained a known zearalenone concentration (information provided by manufacturer), led to somewhat diverse results: spiked samples systematically showed higher values, indicating signal-enhancing matrix effects, whereas for the sample originally containing zearalenone, the concentration obtained from the sensor correlates well with the data provided by the manufacturer (Table 4).

Hence, both sensors introduced by Sergeyeva et al. [68,69] feature good recovery rates for spiked cereal extracts and cereal extracts. Due to their straightforward use and cheap smartphone-based readout system, they are potentially highly interesting for in-field food monitoring, even though they require some sample pre-treatment, i.e., extraction.

To detect fumonisin B1 (FB1)—another mycotoxin—in maize samples, Munawar et al. [70] fabricated nanoMIPs via solid-phase synthesis and immobilized them in a polypyrrole-zinc porphyrin composite for electrochemical sensing via differential pulse voltammetry (DPV) and electrochemical impedance spectroscopy (EIS). Both methods led to an LOD in the femtomolar range, together with high selectivity against five other mycotoxins. Extracts of maize demonstrate a recovery rate of 96–102% in real-life samples.

Zhang et al. [71] also used the “dummy-template” approach to synthesize silica gel MIPs on Mn-doped ZnS quantum dots to detect the mycotoxin patulin via sensing of phosphorescence quenching. The system measures patulin concentrations in PBS buffer in a linear range of 0.43–6.50 µmol/L with LOD = 38.5 µg/L, which is below the maximum allowed limits for fruit juices. The sensor is also selective for patulin, when compared to three other mycotoxins and 5-hydroxymethyl-2-furaldehyde (5-HMF), which is a well-known interferent of patulin detection in apple juice. For their experiments in the matrix, they diluted an apple juice concentrate from 70° Brix to 12° Brix, and spiked the samples with 1–3 µmol/L patulin. They yielded recovery rates of 127.2% ± 5.0% for 1 µmol/L spiking, 110.4% ± 4.4% for 2 µmol/L spiking, and 102.9% ± 3.6% for 3 µmol/L spiking. As these recovery rates were systematically too high, they attributed this to the matrix effects of coexisting interferents in apple juice.

### 3.2. Pesticides

The use of pesticides to enhance harvest and minimize labor costs is an indispensable part of modern agriculture. Nevertheless, such substances are suspected to cause health damage and, therefore, should not end up in food sold to consumers [72]. This generates the need for sensors to detect pesticides at relevant concentrations in food samples. Amatatongchai et al. [73] reported on MIP nanoparticles coupled with magnetite-gold core particles as sensing material for amperometric detection of the carbamate-based pesticide carbofuran (CBF). Optimized measurements in phosphate buffer revealed linear sensor characteristic in a range between 10 nM and 100 µM with an LOD of 1.7 nM and a LOQ of 5.7 nM. Selectivity studies revealed low sensor response to six possible interferents, and no unspecific signal caused by background electrolytes (e.g., NaNO_3_, NaCl). Their measurements in real samples (five types of fruit or vegetables) required sample pre-treatment through homogenization and extraction. The sensors detected the analyte in the spiked samples with a recovery rate between 92.8% and 104.0%. In three of the five unspiked samples, they could detect CBF, which showed that the method was feasible to detect small amounts of the pesticide in real samples. The results were in good agreement with the HPLC reference method.

An earlier paper of the same group [74] introduced an amperometric sensor based on electropolymerized MIPs combined with carbon nanotubes and gold coated magnetite nanoparticles to detect CBF in fruit samples. The sensor exhibited an LOD of 3.8 nM CBF and an LOQ of 12.7 nM CBF. HPLC confirmed the recovery rates of the sensor was between 95.3% and 109.0%.

Last, but not least, Cakir et al. demonstrated MIP nanofilm sensors on quartz crystal microbalance (QCM) and surface plasmon resonance (SPR) transducers [75] to detect the herbicide 2,4-dichlorophenoxyacetic acid (2,4-D) in apples. It revealed a linear response for 2,4-D in a concentration range between 0.23 and 8.0 nM, with LODs of 20.17 ng/L for QCM measurements and 24,57 ng/L for SPR sensing. Only low cross-selectivity could be observed for two structurally related substances (2,4,6-trichlorobenzoic acid and 2,4-dichlorophenol) for both transducers, indicating high selectivity for 2,4-D. For concentration determination in apples the sample components were extracted following the QuEChERS method after homogenization and spiking with 250, 500, or 1000 ng/L 2,4-D. In non-spiked samples, no false positive signal was detected.

### 3.3. Bacteria Contamination

One major advantage of biomimetic receptors is that the concept of molecular imprinting can be transferred to large analytes (e.g., whole cells) [76]. In this context, chemical sensors based on surface-imprinted polymers (SIPs) for detecting bacteria—which are relevant analytes for the monitoring of food hygiene—have been reported and applied in complex food sample matrices. Among those, measurements in apple juice turned out particularly popular in evaluating SIP sensing performances on different transducers [77,78,79]. The work of Cornelis et al. [77], which details the development of a chemical sensor for the fecal indicator *E. coli*, stands out, as they reported detecting their analyte at concentrations below the legal norm in spiked 95% apple juice, without matrix pre-dilution or sample pre-treatments. The sensor was based on a polyurethane SIP layer coupled with heat transfer monitoring as the transducer method. With this, they achieved the detection of *E. coli* in PBS buffer at concentrations below the legal norm for apple juice (1000 CFUs/mL) and interestingly showed that changing the background solution from buffer to apple juice had no negative influence on the sensor performance (Figure 8 and Table 5). Selectivity studies with four other coliform bacteria revealed only low cross-selectivity for these species.

Givanoudi et al. [80] used the same transducer and polymer system to detect *Campylobacter* species (*C. jejuni* and *C. coli*) in chicken cecal droppings, which is a relevant matrix for the detection of these pathogenic food contaminants: chicken meat products are expected to be the main source of campylobacteriosis. For their measurements in a complex matrix, they suspended the chicken cecal droppings in PBS buffer, spiked the suspension with the known bacteria concentration and filtered the sample before sensor measurements (Figure 9). This resulted in low detection limits in PBS buffer and again, the performance of the sensor did not strongly decrease during measurements in a complex matrix (Table 5). To further evaluate the sensor usability for real sample application, they examined the sensor response towards six different, but closely related bacteria species and three different bacteria strains of the template bacteria. Their results revealed “species-selective” behavior, as comparable high cross-selectivity was observed when the sensor was exposed to different strains of the template bacterium, whereas low sensor response was observed for bacteria of different species.

Arreguin-Campos et al. [81] also developed a polyurethane based SIP sensor for the detection of *E. coli* in milk. The receptor performance was tested on two different transducers (heat transfer method and impedance measurements). Even though they achieved a lower detection limit under optimized conditions in buffer during impedance measurements (LOD = 120 CFUs/mL) than with a heat transfer method (LOD = 1070 CFUs/mL), the latter was used for real-sample experiments. For their *E. coli* sensing in milk, they spiked their complex matrix with bacteria, but did not treat the samples any further. The measurements led to a slightly increased detection limit when compared to measurements in PBS buffer, as listed in Table 5.

### 3.4. Food Adulteration

There is also some work that addresses the issue of food adulteration, as illegal blending of products to maximize profit or to lower production costs is a criminal practice that has led to several food scandals during the last decades [82]. In this context, Zeilinger et al. [83] developed a mass-sensitive sensor based on an MIP thin film as sensitive layers to detect melamine in dairy products. The detection of food blending with melamine is a serious issue, not only because the adulterated food pretends to have a higher protein concentration when analyzed with common methods (e.g., Kjeldahl method), but also because melamine is harmfully connected to kidney damage. The developed sensor was reported to measure analyte concentrations to an LOD of 8 µM in proof-of-concept experiments, where only melamine was dissolved in water. The use of dairy products as a complex matrix led to decreased sensor signals due to a complex formation between melamine and matrix proteins. Therefore, the sensor showed its best performance in whey, because it contains comparably little protein. Measurements in milk required 10-fold dilution to obtain useful signals. Even though measurements in a matrix turned out to be complicated, the sensor is still useful in realistic conditions: adulterated food typically shows a high concentration of the illegally blended substance to maximize the effect.

Another application of biomimetic sensors in detecting food adulteration was reported by Cheubong et al. [84]. They developed a sensor based on fluorescence-labelled MIP nanogels, synthesized using their post imprinting method, to detect porcine serum albumin (PSA) as an indicator for pork contamination in halal meat extracts. The sensor was first characterized in measurements without a complex matrix and yielded an LOD of 40 pM and low cross-selectivity towards four other proteins. An evaluation of sensor performance in PSA-spiked beef extract as a matrix revealed the importance of sample dilution: 10 to 100 fold diluted samples yielded significantly reduced signals, whereas the sensor performance for 500 fold diluted beef extract samples was similar to measurements in PBS buffer. In how far those samples indeed constitute a real-life matrix remains an open question, of course. To demonstrate the usability of the sensor to detect meat contamination, they mixed diluted beef extract with diluted pork extract in ranges between 0.01 and 100 wt% pork contamination. Their results revealed a detection limit of 0.1 wt% pork, which was reported to be a better performance than real-time PCR, the established method for this purpose (Figure 10).

## 4. Environmental Monitoring

Environmental monitoring of target analytes such as heavy metals, pesticides, antibiotics, bacteria, or toxins is of great importance as they can be harmful for humans, flora, and fauna. There are numerous publications on MIPs used for their detection, mostly in environmental water samples. These matrices require mostly little pre-treatment. However, they usually contain just trace levels of the respective analyte, which poses a huge challenge to sensors.

Pollution of water samples by heavy metal ions is a serious environmental concern as even ultra-trace amounts of heavy metals can bioaccumulate and affect various biological pathways. Malitesta et al. [85] published a review in 2017 about MIPs used in environmental determination of heavy metals. Di Masi et al. [86] developed ion-imprinted polymer nanoparticles (nanoIIPs) for Cu^2+^ and investigated their application in an electrochemical sensor platform. They synthesized Cu^2+^ nanoIIPs by free radical polymerization and immobilized these on electrodes for electrochemical detection by differential pulse voltammetry (DPV). The sensor showed a higher response for Cu^2+^ nanoIIPs than for non-imprinted polymers, a linear range of 1.9–61 nM, and an LOD and LOQ of 74 pM and 247 pM, respectively, when tested in buffer. It also features good repeatability with an RSD of 3.6% for ten replicates and reproducibility with an RSD of 5.7% for five replicates. Finally, the system is selective against Ni^2+^, Zn^2+^, NO^3−^, and CrO_4_^2−^ (Figure 11). In spiked drinking water samples, the linear range was between 1.9 and 31 nM with an LOD of 0.30 nM and an LOQ of 1.01 nM. The authors claimed that the higher values in spiked water samples compared to standard solutions was due to lower repeatability of measurements in the latter matrix.

Pesticides are commonly used in agriculture to increase the quality and quantity of a crop, which increases the amount of pesticide residues in the environment [87]. Farooq et al. published a review in 2018 on pesticide residue detection in combination with MIPs [88]. In a study in 2020, El-Akaad et al. [89] developed a capacitive sensor based on MIPs for detecting the insecticide imidacloprid. MIP particles were synthesized via photo-initiated emulsion polymerization and immobilized on the sensor surface. They reported an imprinting factor of 6 and selectivity factors of 14.8, 6.8, 7.1 and 8.2 for acetamiprid, clothianidin, thiacloprid, and thiamehoxam, respectively, proving selectivity towards the target analyte. After subtracting the NIP signal from the MIP signal to only register specific interactions, the sensor had a linear range of 5–100 µM and an LOD of 4.61 µM. Furthermore, the system proved reversible and reproducible: it could be regenerated at least 32 times while maintaining a minimum of 90% of the initial system. Measurements in spiked samples gave good recovery rates in the range of 94–106% for six measurements.

Sulfonamides serve as antimicrobial drugs and chemotherapeutic agents, but their extensive use leads to antimicrobial resistance. In addition, they may be allergenic and potentially carcinogenic. Zamora-Galvez et al. [90] synthesized MIP-decorated magnetite nanoparticles (MNP) for specific and label-free detection of sulfamethoxazole (SMX). The magnetic properties of the core material are useful for separation and pre-concentration of the analyte. They measured the amount of captured analyte by electrochemical impedance spectroscopy fixing the MIP-decorated MNPs over the working electrode using a magnet. A linear range of 1 × 10^−2^–1 × 10^−10^ mol/L and an LOD of 1 × 10^−12^ mol/L were found. Furthermore, the sensor had good repeatability and reproducibility with an RSD of 3.6% for five measurements and RSD of 6.8% for three measurements, respectively. After three weeks 82% of the initial response was detected. Furthermore, MIPs showed 3 times higher, and NIPs 1.4 times higher, response compared to the blank measurements (no SMX added). The structural analogues sulfadiazine and sulfacetamide did not show a higher response compared to blank proofing the selectivity of the sensor. Measurements in real seawater samples required some minor pre-treatment: the samples were filtrated to remove potential interfering species and spiked with SMX. Extraction media (methanol acetic acid 1/1, *v*/*v*) was added to extract interfering compounds. The resulting recovery rates were in the range of 87–106% with RSD in the range of 1.2–4.5%.

Hu et al. [91] reported a one-pot synthesis for fluorescent MIPs for sulfapyridine using Mn-doped ZnS quantum dots (QD) as a fluorescent core. The developed fluorescent NPs could be used for sensing sulfapyridine by fluorescent quenching. The FMIP showed a good binding capacity and was highly selective when compared to structural analogues. With increasing SPD concentration, a fluorescent decrease could be monitored in the range of 0–80 µM, with a detection limit of 0.5 µM. Measurements in spiked tab water samples that were used without any pre-treatment led to recovery rates from 95.6% to 99.8% with RSD below 1.4% at three spiking levels.

Cyanobacteria blooms cause a great threat to water bodies, aquatic animals, and human health. Li et al. [92] developed thermosensitive molecularly imprinted core-shell CdTe QD as ratiometric fluorescence nanosensors to detect phycocyanin, which is a fluorescent cyanobacteria-specific pigment protein and an indicator for cyanobacteria. Binding of the analyte to the MIP core-shell QD led to quenching of the QD fluorescent intensity, but at the same time enhanced analyte fluorescence due to fluorescence resonance energy transfer (FRET) from the energy donor of the QDs. This makes a ratiometric sensor feasible. It is not possible to excite phycocyanin in solution due to the larger distance to QDs. Thus, the signals strictly result from the analyte bound by the MIP. At an optimum response time of 10 min a good linearity within the range of 0–1.8 µM with a low detection limit (3.2 nM) was found. The sensor showed high selectivity when compared to phycoerythrin, spirulina powder, and BSA. NIP responses were smaller than the ones for MIPs, leading to imprinting factors of 12.1 and 15.6 for 20 °C and 45 °C, respectively. Stable measurements could be conducted over a 60 min time period. Storage at 4 °C over 6 months without light exposure maintained at least 90% in the initial month. Measurements in filtrated (0.45 µm PTFE syringe filter) and spiked seawater samples showed a recovery of 92.0–106.8% with RSDs of 2.9–5.5% for five measurements.

Sergeyeva et al. [93] synthesized nanostructured polymeric membranes to detect aflatoxin B1 (AFB1), which is one of the most toxic mycotoxins, via fluorescence. The MIPs were highly selective towards AFB1, when compared to the structural analogues aflatoxins B2, aflatoxins G2, and ochratoxin. The detection range of the system turned out to be 14–500 ng/mL. Furthermore, they applied the system to the detection of AFB1 in wastewater from a bread-making plant (Table 6).

Bisphenol A (BPA) is a component that can be found in a wide range of industrial products, such as food containers, electronical products, or medical equipment, as it has been used for the production of polymer components. This wide-spread use reflects itself by BPA contaminations in the environment, which require monitoring, not least due to the toxicologic and endocrinologic action of BPA. Lu et al. [94] developed a detection method synchronously extracting and pre-concentrating BPA onto magnetic molecularly imprinted polymers (BMMIP), with subsequent readout on a magneto actuated glassy carbon electrode (MGCE) by differential pulse voltammetry. This system combines isolation, clean-up, enrichment, and qualification in one step, while immobilization and removal of the magnetic MIP particles on the electrode is controllable by an external magnetic field. The sensor showed excellent binding capacity, short adsorption equilibrium time (30 s), and high selectivity. Furthermore, the BMMIPs showed a good stability in the pH range of 2–8 and a temperature range of 5–45 °C, and a stable binding capacity after five adsorption/desorption cycles, proving reusability. An LOD and LOQ of 0.133 µmol/L and 0.439 µmol/L were determined, respectively. Reproducibility was shown with five successive tests, resulting in an RSD of 0.48%. The sensor system was applied for determination of BPA in food and environmental samples. Tea, municipal sewage, tap water, milk, cabbage, and soil samples were spiked with three different BPA concentrations. Recovery rates ranged from 81.31% to 119.77%, with RSDs ranging from 0.06% to 17.42%. Different sample types required different pre-treatment: Solid samples such as soil and cabbage were ultrasonically extracted with methanol, centrifuged, the supernatants volatilized, and the residues dissolved in methanol and diluted in water. Milk samples were pre-treated with acetonitrile as extraction solvent, the organic phase isolated by centrifugation, and then diluted with water. Lake water, tap water, and tea were directly used after filtration with a hydrophilic membrane.

## 5. Drugs of Abuse and Their Precursors

In their 2018 world drug report, the United Nations Office on Drugs and Crime emphasized the increasing abuse of drugs and non-medical use of prescription drugs. They appeal to the international community to step up and take responsibility for this problem [95]. Detection of these substances is a crucial step to tackle this challenge.

To identify a person intoxicated with an illegal substance, a biological sample such as blood, saliva, or urine has to be taken. In an optimal case, one can analyze the respective target compound directly in the sample matrix without extensive pre-treatment. Smolinska-Kempisty et al. [96] developed a potentiometric sensor based on MIP nanoparticles for cocaine. They obtained so-called nanoMIPs by solid phase synthesis and incorporated them into an ion-selective membrane. Specific recognition and binding of cocaine to the membrane triggered a potential difference across the membrane. Metabolites also bound to the nanoMIPs, however, there was no response for compounds such as alantamine. Furthermore, nanoNIPs did not bind cocaine species either. Measurements in spiked human serum samples (sterile filtered) were possible in a range of 1 nM to 1 mM. The serum matrix did not affect either the linearity or sensitivity of the sensor response. It is not only important to detect the drug itself to prove drug use, but also its metabolites. A recent work [97] reported the development of a fiber-optic sensor for sensing carboxyl-fentanyl, which is the major metabolite of the new designer drug butyrylfentanyl, in human serum. The authors developed a long period fiber grating (LPG) sensor array consisting of two parts: a detection LPG, functionalized with MIPs (synthesized by solid-phase approach), and a reference LPG that takes temperature responses into account. Sensor measurements in the range of 0–1000 ng/mL with the imprinted sensor and a control sensor (functionalized with control polymer particles not specific towards carboxy-fentanyl) showed a certain level of non-specific adsorption to the control polymer, but the MIP sensor yielded substantially higher signals. The minimum concentration detected was 50 nm/mL of carboxyl-fentanyl. When tested for cross-reactivity towards glucose and BSA (abundant in serum), the sensor did not show any response towards glucose and just a minor one to non-specific adsorption of BSA. It even exhibited a greater selectivity to cocaine and morphine and was able to distinguish between the structurally very similar compounds fentanyl and carboxy fentanyl, although there was some response towards fentanyl (30%). Measurements in spiked human serum showed an increase in signal, whereas the blank serum sample did not show a change.

Though the main emphasis of research lies on detecting drugs of abuse in biological matrices, it is also interesting to trace them in the environment: this helps to assess where they are consumed. For instance, De Rycke et al. [98] developed a capacitive sensor to detect benzyl methyl ketone (BMK), an amphetamine precursor, to monitor suspicious amphetamine-type stimulants (ATS) activity. They synthesized MIP nanoparticles and further immobilized these on the sensor surface. The sensor exhibited a linear range of 50 to 100 µM and an LOD of 1 µM in tap water spiked with BMK. The authors reported cross-reactivities of 22%, 17%, and 33% towards amphetamine, *N*,*N*-di-(β-phenylisopropyl)amine, and N-formyl amphetamine, respectively. Furthermore, the sensor worked in a wide pH range (pH 3–12) and showed high stability, repeatability, and reproducibility. However, there were interferences from other ATS markers, which led to false positive results for BMK.

Screening confiscated powders for drugs also constitutes an application scenario for sensors. The work of Florea et al. [99] presented a polymer platform for the electrochemical detection of cocaine in street samples. This task can be challenging, because samples are often adulterated with levamisole, which usually suppresses the response on unmodified electrodes. To reach their goal, the authors electrosynthesized MIPs for cocaine on graphene-modified electrodes. They optimized the sensor in terms of the ideal monomer concentration, number of electropolymerization cycles, scan rate, and the accumulation time of cocaine on the MIP. Under optimized conditions, they reported a linear range of up to 500 µM for pure cocaine and binary (cocaine/levamisole) mixtures. The lowest detectable concentrations were 50 µM for pure cocaine and 100 µM for binary mixtures. Real street samples (containing cocaine and levamisole) were tested by dissolving the samples in PBS buffer (pH 7) and diluting them 1/10. The MIP modified electrodes clearly showed the presence of cocaine, whereas an unmodified electrode led to a false negative result.

In a following study [100], the same group synthesized MIPs for cocaine on graphene-modified electrodes with palladium-integrated nanoparticles. The developed sensor showed a linear range of 100–500 µM, with a detection limit of 50 µM. Moreover, the paper reports good reproducibility with an RSD of 1.89% for three different sensors, and good long term stability at room temperature with 92.6% of the original signal after 28 days. The MIP also proved selective for cocaine when compared to homatropine, which has a similar structure. For real sample analysis, saliva, river water, and street samples were selected. Saliva and river samples were further diluted 1/10 in PBS Buffer pH 7.0 and spiked with cocaine. Cocaine street samples were dissolved in PBS and further 1/1 diluted in PBS. Real sample analysis revealed recovery values of 99.4% (RSD 13.2) and 103% (RSD 6.3%) for saliva and river water, respectively. Furthermore, it turned out possible to detect cocaine in street samples. Although the sensor thus proved feasible to determine cocaine in different complex matrices, its sensitivity is sufficient for applying it in biological samples without pre-concentration.

## 6. Conclusions and Outlook

Overall, research on biomimetic sensors on the basis of MIPs reveals good performance in detecting a high variety of analytes in the presence of complex background matrices. However, when looking further into the details, it becomes clear that many of those systems still require sample preparation steps, while comparably few detect their target analyte in a matrix without such steps. On the other hand, several papers also report strong matrix effects; the only way to suppress those was to dilute samples with a buffer, which raises the question as to how far such systems actually deliver on the claim to produce sensor responses in actual complex environments. Another point is that only a few papers demonstrate measuring actual samples: in most cases, they reveal good recovery rates for real samples spiked with the analyte. Even though many papers indeed demonstrate validation by external reference methods, this does not yet fully prove that the system is fit for the commercial purpose intended; such recovery rates reflect analyte recovery of the sensor measurement itself, but not the entire analytical process containing all necessary sample preparation steps. On the more positive side, the reviewed sensors demonstrate their suitability for applying them in real-world settings, because many of them report detection limits below legal concentration norms or in concentration ranges that were relevant for the corresponding application field. Hence, in terms of the research and its results, the field is developing in a highly appreciable manner. However, there is a reason why most work is still academic and requires further development until potential commercialization: most MIP sensor literature does not discuss batch-by-batch reproducibility of the effects when synthesizing the respective polymer(s) several times. Most intra-assay reproducibility tests reported take place on a given sensor exposed to target analyte concentrations and rely on a limited amount of MIP batches. In contrast to this, application scenarios such as separation or solid phase extraction indeed already see commercialized MIP, so there is no reason to assume that this should not be possible for sensing in principle. One just has to keep in mind that sensing applications require a much higher selectivity of the material than separation, because the analytical process comprises of one equilibrium/steady state rather than many, as is the case, e.g., in chromatographic columns. Nonetheless, the imprinting field is indeed developing to a more mature stage where, on the one hand, it is still important to gain deeper understanding of the physical and chemical effects underlying recognition; but on the other hand, the last years have indeed seen a strong drive towards developing actual sensors and assay systems and, thus, carry such biomimetic receptor strategies further towards commercial use. Given the regulatory background and competition by other methods, MIP-based environmental monitoring, in our opinion, seems closest to the market due to less complex sample matrices and less complex legislation, than in the case of medical or food safety applications. Future research will have to focus on more uniform binding properties, especially in the case of MIP thin films; when synthesizing MIP nanoparticles, it is already possible to achieve “monoclonal polymer antibodies” [3] with a very narrow affinity distribution. However, one achieves this by selectively eluting high-affinity particles during solid phase synthesis and discarding those showing weaker binding. This also clearly demonstrates that we still do not fully understand the phenomena leading to imprints in a strict, quantitative manner.

## Figures and Tables

**Figure 1 sensors-21-05550-f001:**
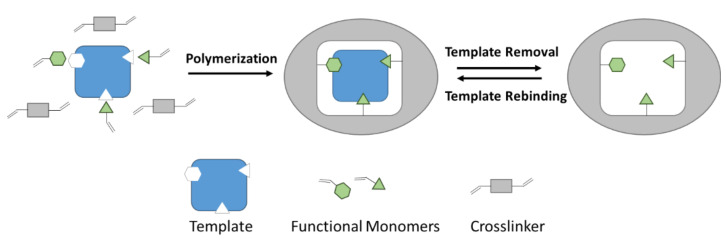
Schematic illustration of imprinting and rebinding.

**Figure 2 sensors-21-05550-f002:**
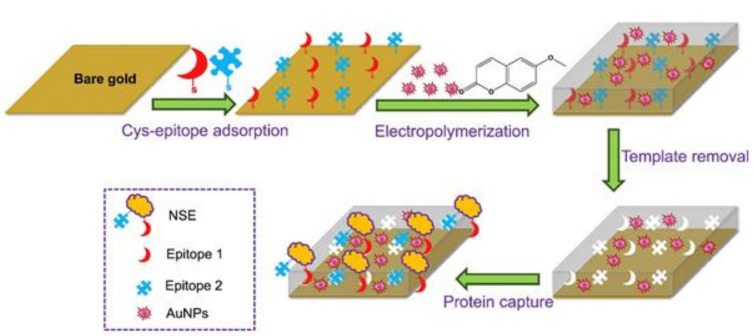
The principle of protein detection using the AuNP-decorated hybrid MIP sensor, which was fabricated by electropolymerization technique in the presence of two different NSE epitopes and AuNPs, reprinted with permission from [28], © Elsevier.

**Figure 3 sensors-21-05550-f003:**
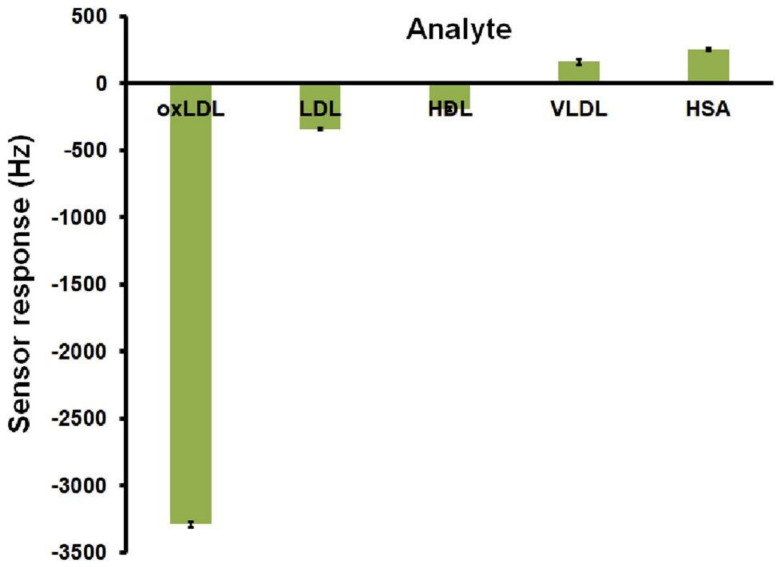
QCM signals obtained from selectivity measurements of the oxLDL-MIP in the presence of various (lipo) proteins at 200 mg/mL, reprinted with permission from [34], © Elsevier.

**Figure 4 sensors-21-05550-f004:**
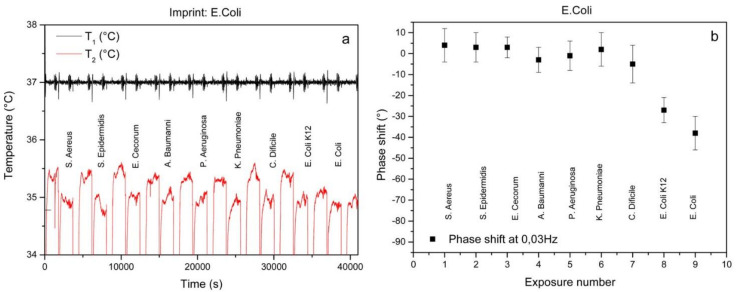
Cross-selectivity experiment showing the time-dependent temperature data for an *E. coli* SIP exposed to eight competitor bacteria and the target consecutively (**a**). The TWTA at 0.03 Hz is also shown (**b**). Both the temperature profile and TWTA data show some degree of cross-selectivity between the two *E. coli* strains [52].

**Figure 5 sensors-21-05550-f005:**
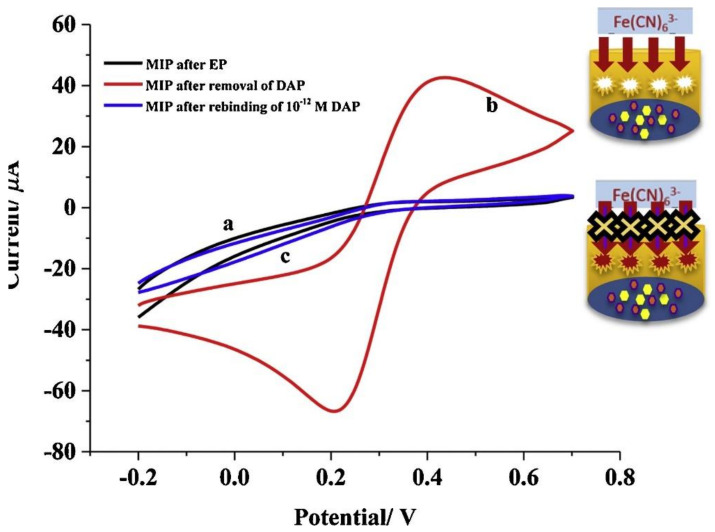
Cyclic voltammograms of (**a**) MIP/Au-Pt NPs/GCE after MIP synthesis (**b**) after removal of DAP (**c**) after rebinding of 1 pM DAP, 5 mM [Fe(CN)_6_]^3−/4−^ mixture solution in 100 mM KCl scanning between −0.2 and 0.8 V at a scan rate of 50 mV/s. Reprinted with permission from [57], © Elsevier.

**Figure 6 sensors-21-05550-f006:**
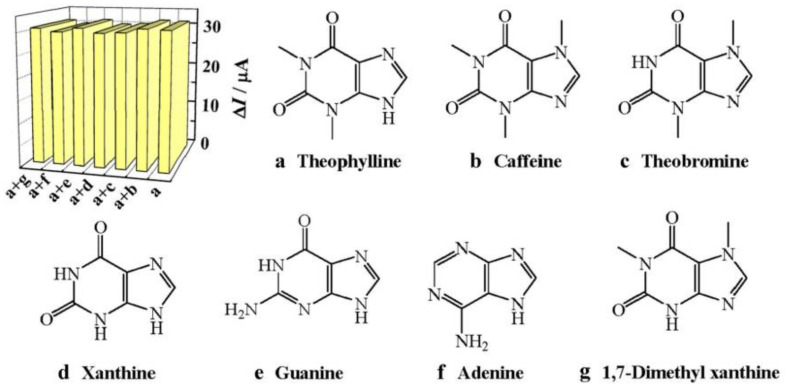
DPV responses on SiO_2_@TiO_2_@MIP/CPE for 1.0 μM theophylline present in binary mixture (1:5) with structural analogs, reprinted with permission from [60], © Springer.

**Figure 7 sensors-21-05550-f007:**
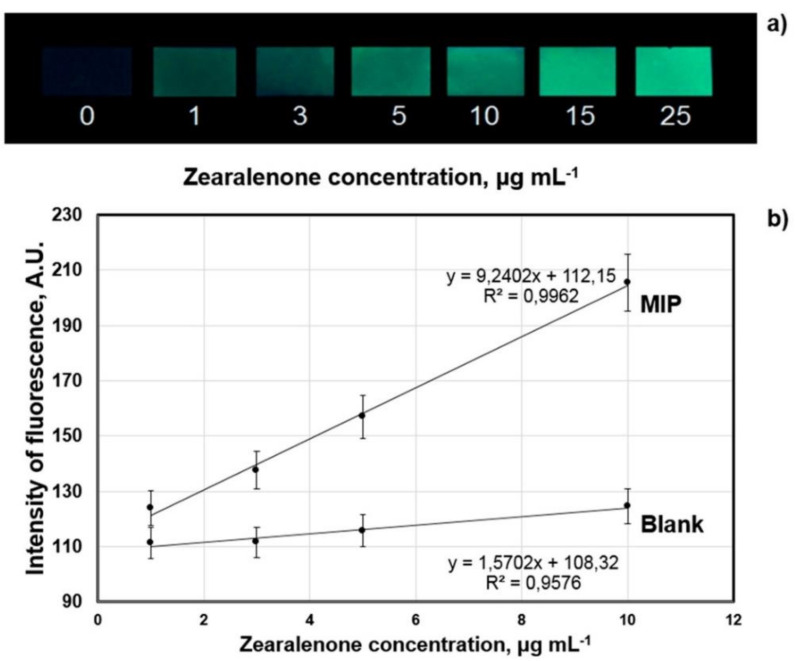
MIP Sensor fluorescence after incubation in zearalenone (1–25 µg/mL) solutions (**a**); Typical calibration curve of the sensor with smartphone readout system for zearalenone detection (**b**); [69].

**Figure 8 sensors-21-05550-f008:**
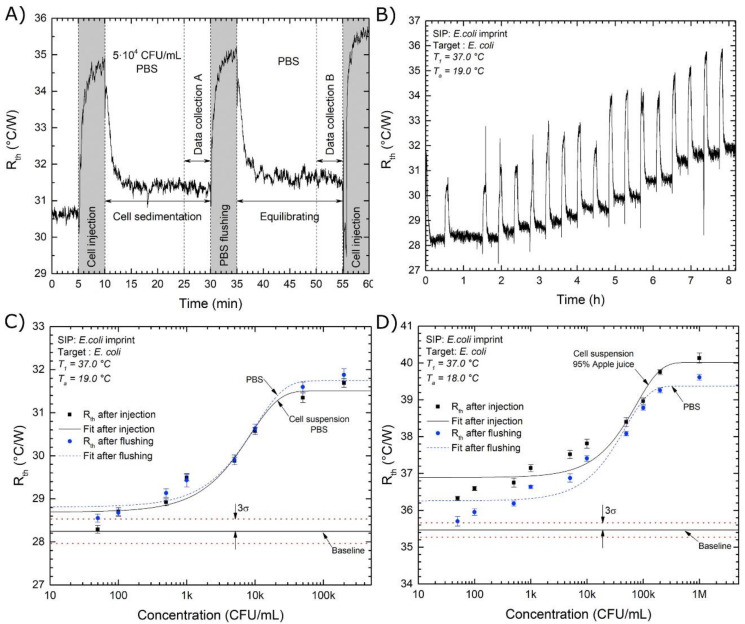
(**A**) Typical four-step exposure protocol (**B**) Dose-response experiment for *E. coli* sensing. The sensor was exposed to increasing bacteria concentrations in PBS buffer (50, 100, 500, 1 × 10^3^, 5 × 10^3^, 1 × 10^4^, 5 × 10^4^, and 2 × 10^5^ CFU/mL). Sensor was flushed with PBS buffer after each injection. (**C**,**D**) Dose-response curves for *E. coli* sensing in PBS and 95% apple juice. The LOD (3σ of baseline measurement) is marked by the red dotted line; Reproduced with permission from [77], © Elsevier.

**Figure 9 sensors-21-05550-f009:**
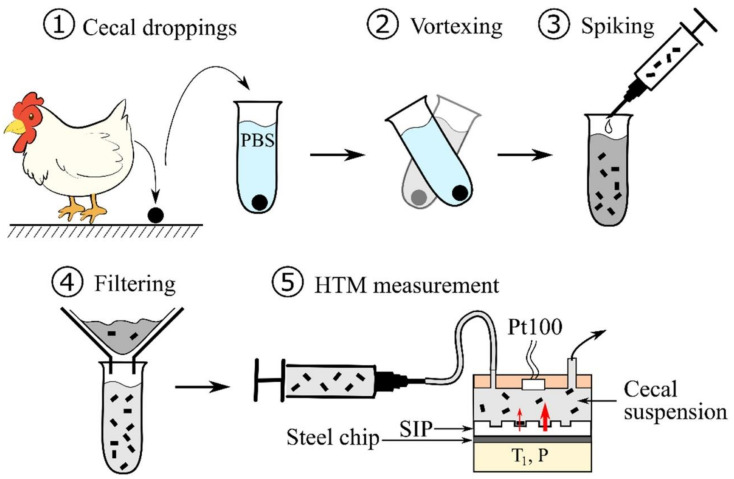
Schematic workflow of sample pre-treatment for *Campylobacter* detection in cecal dropping extracts with SIP-based HTM measurements; Reproduced with permission from [80], © Elsevier.

**Figure 10 sensors-21-05550-f010:**
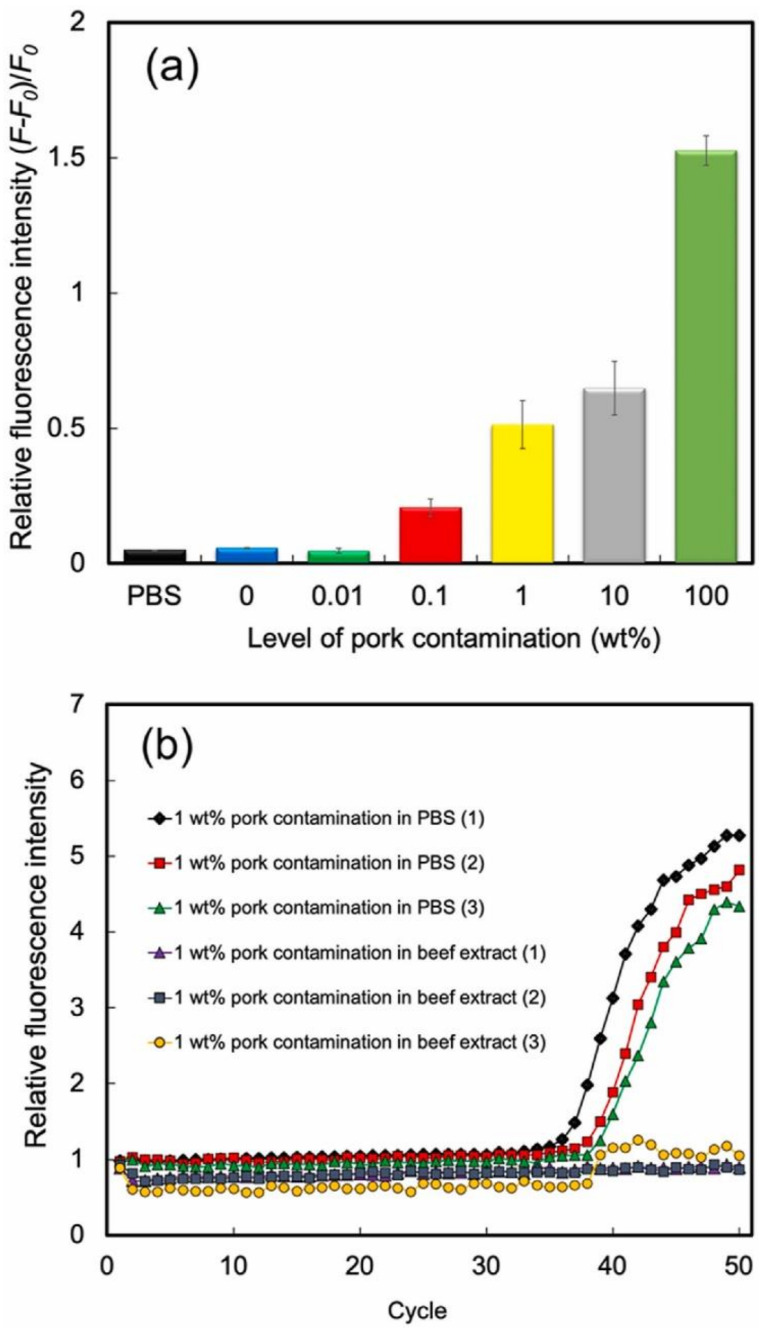
Sensor signal for pork contamination in beef extract samples (0–100 wt%), *n* = 3 (**a**). The real-time PCR results of 1 wt% pork contamination in PBS and beef extract samples, *n* = 3 (**b**); Reproduced with permission from [84], © Elsevier.

**Figure 11 sensors-21-05550-f011:**
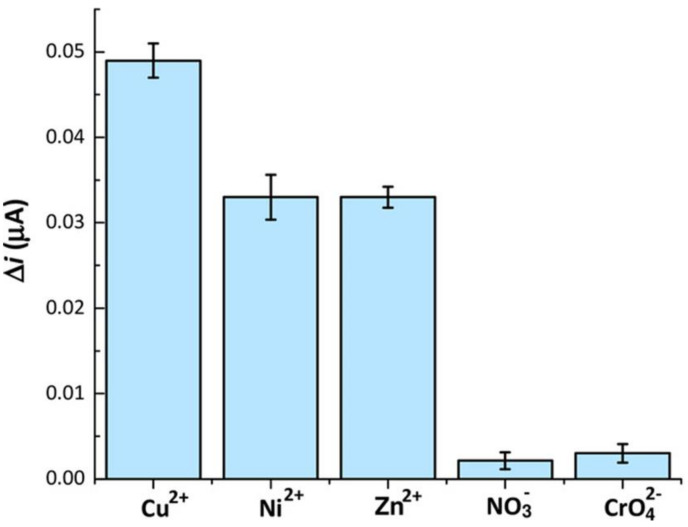
Selectivity studied of nanoIIPs to 1.9 nM of Cu^2+^, Ni^2+^, Zn^2+^, NO^3−^ and CrO_4_^2−^. Reproduced with permission from [86], © Wiley-VCH.

**Table 1 sensors-21-05550-t001:** A selection of target biomarkers for MIP-based sensors and their characteristics.

Biomarker	Linear Range	LOD	Complex Matrix	Recovery Rate [%]	Reference
Lysozyme	150 nM–20 µM (PBS)	141 nM (PBS)	Synthetic saliva	93–113	[37]
Creatinine	0.5–150 μM (PBS)	0.077 μM (PBS)	Human serum and urine	1.47–5.55% relative difference to reference method	[38]
8-hydroxy-2′-deoxyguanosine	0.005–50 μM (PBS)	0.0008 µM (PBS)	Human serum and urine	94–107	[38]
Serotonin	0.01–1000 µM (PBS)	0.002 µM (PBS)	urine	102–111	[39]
Heart fatty acid binding protein	-	4.18 ng/mL (PBS)	Fetal bovine serum	-	[40]
ST2	-	8.79 ng/mL (PBS)	Fetal bovine serum	-	[40]
IL-6	0.1 pg/mL (PBS)	< 0.1 pg/mL (PBS)	Human serum	-	[41]
IL-2	35 fg/mL–39 pg/mL (synthetic human serum)	5.91 fg/mL	Synthetic human serum	-	[42]
L-hydroxyproline	0.2–100 ng/mL (borate buffer)	0.05 ng/mL (borate buffer)	Human serum	97.9–102.53	[43]
NS 1 (Dengue Fever biomarker)	1–200 ng/mL (PBS)	0.3 ng/mL (PBS)	Human serum	95–97.14	[44]
Fibrinopeptide B	0.2–22 ng/mL (urine)	0.13 ng/mL (urine)	urine	-	[45]
CA-125	0.01–500 U/mL (PBS)	U/mL (PBS)	Synthetic human serum	91–105	[46]

**Table 2 sensors-21-05550-t002:** A selection of target analytes for MIP-based sensors aiming at therapeutic drug monitoring.

Drug	Linear Range	LOD	Complex Matrix	Recovery Rate [%]	Reference
metronidazole	50–1200 ng/mL (PBS)	17.4 ng/mL (PBS)	Human serum	93.5–102.7	[58]
velpatasvir	0.649–80.0 ng/mL (phosphate buffer)	0.21 ng/mL (phosphate buffer)	Plasma and urine	101.2–101.5	[59]
insulin	50–2000 pM (plasma)	0.081 pM	plasma	99.5–101	[61]
puerarin	0.02–40 µM	0.006 µM	urine	96.7–102.9	[62]
cytarabine	1.0 × 10^−6^–1.0 × 10^−3^ M (phosphate buffer)	5.5 × 10^−7^ (phosphate buffer)	Human serum and urine	98.6–99.1 (urine) 95.3–103 (serum)	[63]
azithromycin	13.33 nM–66.66 µM	0.85 nM	Plasma, urine and tears	102.4 (plasma) 98.03–106.33 (urine) 99.06–109.88 (tears)	[64]
oxaliplatin	1.0–20.0 nM 20.0–250.0 nM	0.39 nM (serum) 0.41 nM (urine)	Human serum and urine	95.7–106.9 (serum) 94.6–108.3 (urine)	[65]

**Table 3 sensors-21-05550-t003:** AFB1 detection in wheat and maize flour samples; Sample 1 was maize extract originally containing AFB1 whereas samples 2–5 were spiked samples; *n* = 5 in all experiments; Reproduced with permission from [68], © Elsevier.

Sample No.	Amount of AFB1 in Sample	Amount of AFB1 Measured with Sensor	Recovery [%]
1	7 µg kg^−1^	6.1 ± 0.52 µg kg^−1^	87 ± 7
2	10 ng mL^−1^	9.6 ± 1.31 ng mL^−1^	96 ± 13
3	50 ng mL^−1^	48.7 ± 1.21 ng mL^−1^	97 ± 2
4	80 ng mL^−1^	76.8 ± 14.69 ng mL^−1^	96 ± 18
5	100 ng mL^−1^	96.3 ± 8.17 ng mL^−1^	96 ± 8

**Table 4 sensors-21-05550-t004:** Zearalenone detection in cereal samples using MIP membrane-based fluorescence sensor; *n* = 5 in all experiments; [69].

Sample	Amount of Zearalenone in the Sample	Amount of Zearalenone Determined with Sensor
maize flour “Dobrodiya Foods”, Kyiv, Ukraine	1 µg mL^−1^	1.9 ± 0.4 µg mL^−1^
wheat flour “Kyivmlyn”, Kyiv, Ukraine	3 µg mL^−1^	4 ± 0.5 µg mL^−1^
rye flour “Dobrodiya Foods”, Kyiv, Ukraine	5 µg mL^−1^	5 ± 0.5 µg mL^−1^
Romer Labs-Check-Sample-Survey CSSMY012-M17161DZ	114 µg kg^−1^	113.2 ± 7.8 µg kg^−1^

**Table 5 sensors-21-05550-t005:** Comparing different SIP-based bacteria sensors.

Detected Bacterium	Complex Matrix	LOD in PBS Buffer [CFUs/mL]	LOD in Matrix [CFUs/mL]	Reference
*E. coli*	95% apple juice	100	100	[77]
*S. paratyphi*	apple juice 10× diluted in PBS	1.4 × 10^6^	2.5 × 10^6^ *	[78]
*E. coli*	apple juice 10× diluted in PBS	70	100 *	[79]
*C. coli*	chicken cecal droppings extract in PBS	1.0 × 10^3^	1.1 × 10^3^	[80]
*C. jejuni*	chicken cecal droppings extract in PBS	6.6 × 10^3^	2.7 × 10^4^	[80]
*E. coli*	milk	1.07 × 10^3^	2.60 × 10^3^	[81]

* lowest measured concentration was taken as LOD.

**Table 6 sensors-21-05550-t006:** Detection of aflatoxin B1 in waste water from a bread-making plant (Kiev, Ukraine) spiked with AFB1, *n* = 3, P = 0.95. Reproduced with permission from [93], © Elsevier.

No.	AFB1, Added to the Waste Water Sample [ng/mL]	AFB1, Detected in the Waste Water Sample [ng/mL]
1	50	55 ± 6
2	100	110 ± 12
3	200	285 ± 25

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
