# Peer review of "Biomimetic Sensors to Detect Bioanalytes in Real-Life Samples Using Molecularly Imprinted Polymers: A Review"

_sensors, 2021, doi:10.3390/s21165550_

Round 1
Reviewer 1 Report
The review paper presents a broad range of studies related to the detection of bioanalytes in biologically relevant matrices, presenting the findings in a very concise and well written format. A few minor tweaks and additions are suggested to ensure that the document covers the full spectrum of relevant research currently available:
Inconsistent capitalization of compounds e.g. line 303: “catechol”, line 326 “daptomycin”
Line 238: “Aside of viruses” should be “Besides viruses, bacteria are also of fundamental interest in sensing:…”
Table 5 compares the detection of bacterium in different complex matrices, only 3 examples are given. This should be expanded upon, as there are more SIP based sensory formats that have been used for the detection of bacteria in complex real world samples. E.g.
- R. Arreguin-Campos et al “Biomimetic sensing of Escherichia coli at the solid-liquid interface: From surface-imprinted polymer synthesis toward real sample sensing in food safety” – measured in Milk
- Zhao, X.; Cui, Y.; Wang, J. Preparation of fluorescent molecularly imprinted polymers via pickering emulsion interfaces and the application for visual sensing analysis of Listeria Monocytogenes. Polymers 2019, 11, 984. – milk
- Perçin, I.; Idil, N.; Bakhshpour, M.; Yılmaz, E.; Mattiasson, B.; Denizli, A. Microcontact imprinted plasmonic nanosensors: Powerful tools in the detection of salmonella paratyphi. Sensors 2017, 17, 1375. – apple juice
Section 5 on drugs of abuse should be extended towards MIP-based dye displacement assays as there has been some research conducted towards the colorimetric detection of illicit substances.
Author Response
The review paper presents a broad range of studies related to the detection of bioanalytes in biologically relevant matrices, presenting the findings in a very concise and well written format. A few minor tweaks and additions are suggested to ensure that the document covers the full spectrum of relevant research currently available:
We very much appreciate the reviewer’s overall positive assessment and the constructive feedback.
- Inconsistent capitalization of compounds e.g. line 303: "catechol", line 326 "daptomycin"
We chose not to capitalize compounds and adjusted the manuscript accordingly.
- Line 238: "Aside of viruses" should be "Besides viruses, bacteria are also of fundamental interest in sensing:..."
Manuscript adapted accordingly.
- Table 5 compares the detection of bacterium in different complex matrices, only 3 examples are given. This should be expanded upon, as there are more SIP based sensory formats that have been used for the detection of bacteria in complex real world samples. E.g.
- Arreguin-Campos et al "Biomimetic sensing of Escherichia coli at the solid-liquid interface: From surface-imprinted polymer synthesis toward real sample sensing in food safety" – measured in Milk
- Zhao, X.; Cui, Y.; Wang, J. Preparation of fluorescent molecularly imprinted polymers via pickering emulsion interfaces and the application for visual sensing analysis of Listeria Monocytogenes. Polymers 2019, 11, 984. – milk
- Perçin, I.; Idil, N.; Bakhshpour, M.; Yılmaz, E.; Mattiasson, B.; Denizli, A. Microcontact imprinted plasmonic nanosensors: Powerful tools in the detection of salmonella paratyphi. Sensors 2017, 17, 1375. – apple juice
We extended Table 5 and also added a new paragraph, discussing the paper of Arreguin-Campos et al. We did not include the paper of Zhao et al. as most experiments in the paper were based on plate counting and fluorimetric sensing of bacteria; binding in real sample was only qualitatively evaluated. Therefore, they could not provide an LOD.
- Section 5 on drugs of abuse should be extended towards MIP-based dye displacement assays as there has been some research conducted towards the colorimetric detection of illicit substances.
Dye displacement assays were left out intentionally as the focus was set on (label-free) sensor systems. None of the other chapters discusses assay formats.
Reviewer 2 Report
The manuscript is a good overview of molecular imprinting for sensing applications including biosensing, food safety, and environmental contaminants, and provides a useful review both for those in the field and newcomers. I recommend accepting the paper with some minor corrections:
1) The abstract (and conclusion) claims that MIPs are not mature enough for commercialisation. I'm aware of some commercialisation activity around this, and the statement is possibly not relevant in the context of a review article.
2) There are some inconsistencies with units, e.g. on line 83.
3) Line 111 introduces the term "anti-CD9" without prior definition, then CD9 is defined later in line 115. This section should be rewritten slightly to provide more clarity.
4) Figure 3 should state in the description that it's a QCM measurement for clarity.
5) There are a few typos that are quite striking - on lines 255 and 287, for instance.
6) Line 411 describes a known concentration of zearalenone - how was this known? Did the manufacturer test the batch, or spike it for the study?
7) Line 477 - QuEChERS, surely?
Author Response
The manuscript is a good overview of molecular imprinting for sensing applications including biosensing, food safety, and environmental contaminants, and provides a useful review both for those in the field and newcomers. I recommend accepting the paper with some minor corrections:
We very much appreciate the reviewer’s overall positive assessment and the constructive feedback.
- The abstract (and conclusion) claims that MIPs are not mature enough for commercialisation. I'm aware of some commercialisation activity around this, and the statement is possibly not relevant in the context of a review article.
We agree that our statement has not been specific enough. Indeed, there are commercialized MIP systems, especially in solid phase extraction and liquid chromatography. However, this is not the case of sensing. Therefore, we amended both the abstract and the conclusions to make this clearer and to give a possible outlook on this question. However, we do not want to delete the statement entirely, not least to demonstrate that MIP still need to deliver on the claim of commercializable low-cost sensing.
- There are some inconsistencies with units, e.g. on line 83.
The inconsistencies were corrected.
- Line 111 introduces the term "anti-CD9" without prior definition, then CD9 is defined later in line 115. This section should be rewritten slightly to provide more clarity.
The section was rewritten accordingly.
- Figure 3 should state in the description that it's a QCM measurement for clarity.
The figure caption was adjusted accordingly.
- There are a few typos that are quite striking - on lines 255 and 287, for instance.
Typos were corrected.
- Line 411 describes a known concentration of zearalenone - how was this known? Did the manufacturer test the batch, or spike it for the study?
Adapted according to remark. The known concentration of zearalenone was provided by the manufacturer.
- Line 477 - QuEChERS, surely?
We corrected the typo.
Reviewer 3 Report
The aim of the review paper was declare the status of the MIP sensors in real sample applications. The review paper is divided in to 4 major chapters: 1. Medical applications; 2. Food Safety; 3. Environmental Monitoring; 4. Drugs of Abuse and their Precursors.
Some attention on the citation should be paid. The name of author in reference [5] should be corrected. The literature or references in table 5 should be listed according to the unified style. The overall impression of the list of references is good. The list of references is correct and appropriate.
The quality of figures should be improved. The font size of the text in figures should be increased. Now the text is barely readable: fig. 1, fig. 2, fig. 4, fig. 5, fig.6, again fig. 5, again fig. 6, fig. 8.
Some numbers of figures are repeated (fig. 5, fig.6, then again fig. 5, and again fig. 6, fig. 11 after fig. 8). The figures should be renumbered and their appropriate designation in the text should be corrected.
The table of content could help the reader navigate through the article and the list of abbreviations could help understand the text without searching the meaning.
Section of “4.2. Pesticides” seems redundant, because such name already is of the section “3.2. Pesticides”. Maybe these two sections could be joined without loosing the quality of paper?
A chapter should contain more references neither a single one (4.1. Heavy Metals) or this chapter should be joined to another chapter.
Author Response
The aim of the review paper was declare the status of the MIP sensors in real sample applications. The review paper is divided in to 4 major chapters: 1. Medical applications; 2. Food Safety; 3. Environmental Monitoring; 4. Drugs of Abuse and their Precursors.
We very much appreciate the reviewer’s overall positive assessment and the constructive feedback.
- Some attention on the citation should be paid. The name of author in reference [5] should be corrected. The literature or references in table 5 should be listed according to the unified style. The overall impression of the list of references is good. The list of references is correct and appropriate.
Table 5 was adapted.
- The quality of figures should be improved. The font size of the text in figures should be increased. Now the text is barely readable: fig. 1, fig. 2, fig. 4, fig. 5, fig.6, again fig. 5, again fig. 6, fig. 8.
The figure quality was improved and figure sizes were increased.
- Some numbers of figures are repeated (fig. 5, fig.6, then again fig. 5, and again fig. 6, fig. 11 after fig. 8). The figures should be renumbered and their appropriate designation in the text should be corrected.
Figure numbering was corrected in the captions and the text.
- The table of content could help the reader navigate through the article and the list of abbreviations could help understand the text without searching the meaning.
A table of contents and a list of abbreviations were added to the manuscript.
- Section of "4.2. Pesticides" seems redundant, because such name already is of the section "3.2. Pesticides". Maybe these two sections could be joined without loosing the quality of paper?
Subchapters of chapter 4 were removed.
- A chapter should contain more references neither a single one (4.1. Heavy Metals) or this chapter should be joined to another chapter.
An additional reference was added. Furthermore, the subchapters of chapter 4 were removed.
Reviewer 4 Report
The manuscript submitted to Sensors entitled "Biomimetic Sensors to detect Bioanalytes in Real-Life Samples using Molecularly Imprinted Polymers: A review" by Birgit Bräuer and co-workers presents a concise short review with an adequate number of references and information. The clear majority of relevant literature about this subject is referenced and properly discussed by the authors. This is a relevant topic with an increasing significance in recent years, carefully prepared, concise, and is a welcome addition to the literature. Therefore, this short revision has enough novelty/importance and should be considered for publication.
However, the conclusion would benefit if the authors discuss the future research directions of this area. Furthermore, considering the major diferences the biomedical, food and environmental sensors and the sensing matrixes of each one, which ones the authors feel have more potential to reach the market first?
Author Response
The manuscript submitted to Sensors entitled "Biomimetic Sensors to detect Bioanalytes in Real-Life Samples using Molecularly Imprinted Polymers: A review" by Birgit Bräuer and co-workers presents a concise short review with an adequate number of references and information. The clear majority of relevant literature about this subject is referenced and properly discussed by the authors. This is a relevant topic with an increasing significance in recent years, carefully prepared, concise, and is a welcome addition to the literature. Therefore, this short revision has enough novelty/importance and should be considered for publication.
We very much appreciate the reviewer’s overall positive assessment and the constructive feedback.
- However, the conclusion would benefit if the authors discuss the future research directions of this area. Furthermore, considering the major differences the biomedical, food and environmental sensors and the sensing matrixes of each one, which ones the authors feel have more potential to reach the market first?
This is indeed an interesting question, especially in the context of a review. We are aware that following these remarks necessarily introduces a pinch of personal opinion and speculation into the article, which is valid in the context of an outlook. Therefore, we amended the conclusions and renamed the section “Conclusions and Outlook”.